*The Company of*
**Biologists**

# A protocadherin mediates oral placode morphogenesis in the tunicate *Ciona*

Sriikhar Vedurupaka[‡], Bita Jadali[‡], Christopher J. Johnson, Alberto Stolfi[§] and Sydney Popsuj*[,§]

## ABSTRACT

In chordate embryos, placodes are ectodermal thickenings around the borders of the neural plate that give rise to various sensory organs and cell types. While generally thought to be a vertebrate-specific innovation, homologous placodes are proposed to exist in non-vertebrate chordates as well. In *Ciona robusta*, a solitary tunicate, the adult mouth (the oral siphon) is derived from one such 'cranial-like' placode in the larva, which we term the oral siphon placode (OSP). At embryonic and larval stages, the OSP consists of a small rosette of cells that form from the neuropore at the anteriormost extent of neural tube closure. While the morphogenesis of the OSP and its physical separation from other surface ectoderm structures have been described in detail, how this is regulated at the molecular level is currently unknown. Here, we show the involvement of protocadherin-mediated cell-cell interactions in the proper morphogenesis of the OSP. *Protocadherin.e* (*Pcdh.e*) is expressed specifically in the OSP, but not in other surface ectoderm cells. CRISPR/Cas9-mediated disruption of *Pcdh.e* in these cells results in the failure of the OSP to physically separate from other structures derived from the same cell lineage. Overexpression of *Pcdh.e* throughout the anterior surface ectoderm results in a similar loss of a physically separate and distinct OSP territory. This effect is likely mediated by homophilic adhesion *in trans*, as Pcdh.e with scrambled extracellular domains failed to recapitulate the phenotype. Finally, we show that *Pcdh.e* expression in the OSP depends on oral placode-specific transcription factors such as Six1/2 and Pitx. Our results suggest that OSP morphogenesis requires precise regulation of a homotypic cell-cell adhesion molecule, which might reflect a conserved mechanism for placode formation in chordates.

KEY WORDS: ***Ciona*, Tunicate, Protocadherin, Placode, Six1/2, Pitx**

## INTRODUCTION

Tunicates are the closest living relatives to vertebrates, and their simple embryos and compact genomes make them attractive model organisms to study chordate evolution and vertebrate origins (Fodor et al., 2021). The oral siphon of tunicate *Ciona robusta* is widely

School of Biological Sciences, Georgia Institute of Technology, Atlanta, GA 30332, USA.
*Present address: Biology Department, Swarthmore College, Swarthmore, PA 19081, USA.
‡These authors contributed equally to this work

§Authors for correspondence (spopsuj1@swarthmore.edu; alberto.stolfi@biosci.gatech.edu)

 A.S., 0000-0001-7179-9700; S.P., 0000-0001-8346-7623

accepted as homologous to the vertebrate mouth and originates from the incomplete closure of the anterior neuropore (Veeman et al., 2010). As the neural plate rolls up from posterior to anterior in a coordinated 'zippering' process (Hashimoto et al., 2015), the anterior end of the neural tube remains open and gives rise to the stomodeum, or oral siphon placode (OSP) (Veeman et al., 2010). After metamorphosis, this placode will give rise to the exterior opening of the mouth, also called the oral or incurrent siphon, of the post-metamorphic juvenile and later adult. Interestingly, the oral siphon of adult tunicates possesses hair cell-like sensory cells (Manni et al., 2004a). Thus, the OSP might be similar to cranial placodes that give rise to various sensory cells and ganglia in vertebrates (Anselmi et al., 2024; Fritzsch and Glover, 2024; Manni et al., 2004b). Like vertebrate cranial placodes, the OSP is a thickening of the surface ectoderm arising from the anterior border of the neural plate, forming a small rosette of tightly clustered cells that are distinct from the surrounding epidermis (Fig. 1A) (Manni et al., 2005; Veeman et al., 2010). However, how these cells form a distinctive cluster, to the exclusion of surrounding surface ectoderm, is not known. It has been shown that homotypic cell adhesion molecules are crucial for neural tube closure (Hashimoto and Munro, 2019; Smith et al., 2021), but how such molecules affect OSP development has not been fully investigated. Interestingly, several conserved placodal regulatory genes such as *Six1/2* (Schlosser, 2006; Schlosser et al., 2008) have been described in the cells, which produce the OSP within larval development. These expressional patterns pushed us to further investigate candidate genes associated with the maintenance of vertebrate placodes in *Ciona*, such as protocadherins, cell adhesion molecules that represent the largest group within the cadherin superfamily (Morishita and Yagi, 2007).

In *Ciona*, protocadherins are represented by five different genes with dynamic expression patterns during embryogenesis, as part of a larger family of classical cadherin and cadherin-related genes (Noda and Satoh, 2008). One of these, which we termed *Protocadherin.e* (*Pcdh.e*; KyotoHoya gene ID KH.C9.518, previously referred to as *Ci-δ-protocadherin-5*), was shown to be expressed in the *Ciona* OSP (Fig. 1B) as the cells begin to coalesce and separate from other surface ectoderm cells (Fig. 1C) (Gibboney et al., 2020; Noda and Satoh, 2008). Of note, it is the only cadherin or protocadherin gene that is expressed exclusively in and around the OSP and not in other surface ectoderm cells (Noda and Satoh, 2008). In vertebrates, placode rosette formation is driven in part by the formation of homotypic cadherin-dependent apical adherens junctions (Breau and Schneider-Maunoury, 2015). Due to the highly localized, specific expression of *Pcdh.e* in the OSP, we hypothesized that this adhesion protein-encoding gene (Fig. 1D) plays an analogous role in mediating homotypic adhesion between the cells in the OSP, allowing them to set themselves aside as a self-organizing rosette. Here, we show through a combination of *Pcdh.e* overexpression and CRISPR/Cas9-mediated disruption that the proper segregation of OSP progenitors as a single, cohesive unit that is physically separate

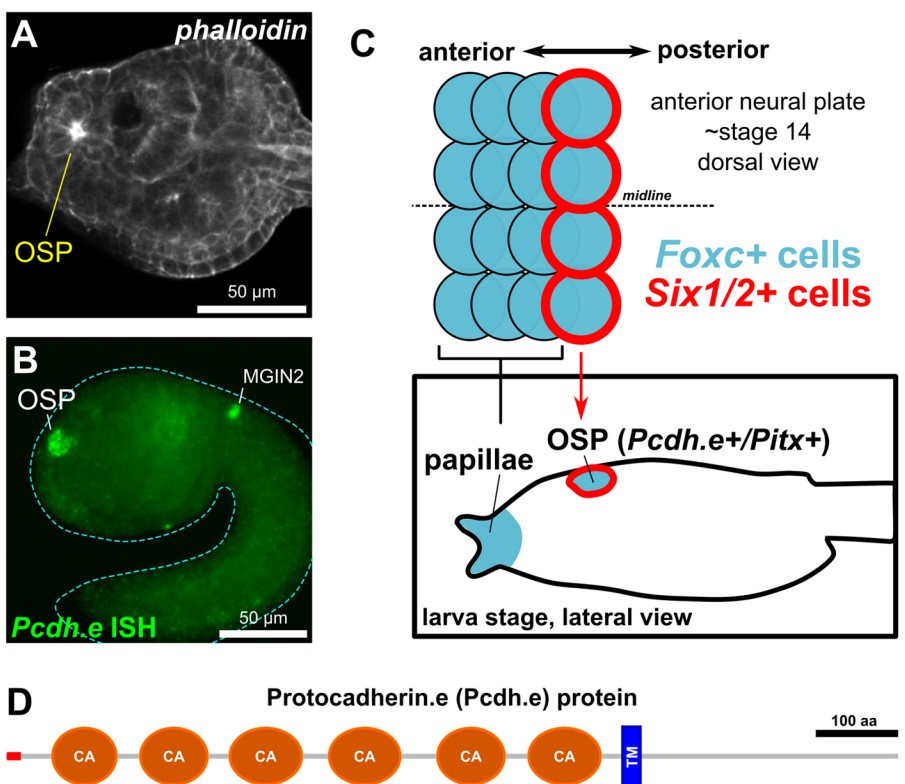

**Fig. 1 *Protocadherin.e (Pcdh.e)* is expressed in oral siphon placode (OSP) cells.** (A) *Ciona robusta* (*intestinalis* type A) embryo at Hotta stage 25 (late tailbud III, ~13 h postfertilization at 20°C), showing cell outlines and the OSP rosette revealed by phalloidin-Alexa Fluor 546 staining. (B) *In situ* mRNA hybridization revealing *Pcdh.e* expression in the presumptive OSP, as well as in the motor ganglion interneuron 2 (MGIN2) cells as previously reported (Gibboney et al., 2020). (C) Summary diagram of the separation of Six1/2+, future *Pcdh.e*- and *Pitx*-expressing OSP cells (red outline) from other Foxc+ cells (blue), mainly those contributing to the papillae. (D) Protein domain analysis diagram of Pcdh.e from SMART (Letunic et al., 2021), showing the presence of a signal peptide (red block), six extracellular cadherin repeats (CA), and a transmembrane (TM) domain, similar to the organization of vertebrate protocadherin family proteins. aa, amino acids. Scale bars: 50 µm.

from other cells in the same cell lineage likely depends on homotypic Pcdh.e-mediated adhesion, linking its highly localized expression to the formation of this important, potentially conserved sensory placode and oral opening in a non-vertebrate chordate.

## RESULTS

### Tissue-specific disruption of *Pcdh.e* by CRISPR/Cas9

*Pcdh.e* expression was previously shown by *in situ* mRNA hybridization to be expressed in cells of the OSP at tailbud stages, the only protocadherin gene to show such OSP-specific expression (Gibboney et al., 2020; Noda and Satoh, 2008) (Fig. 1B). We therefore decided to test its function in the OSP using tissue-specific CRISPR/Cas9-mediated mutagenesis as previously described (Stolfi et al., 2014). We targeted the *Pcdh.e* gene using a combination of two validated single-chain guide RNAs (sgRNAs) predicted to target exons 2 and 3 (Fig. S1A). Quantification of sgRNA-induced indels by next-generation sequencing revealed an efficacy of at least ~20% and ~30%, respectively (Fig. S1A). To restrict Cas9 activity to the lineage leading to the OSP progenitors, we used the *Friend of GATA* (*FOG*) promoter to drive its expression (*FOG>Cas9*). Embryos were co-electroporated with sgRNA and Cas9 expression plasmids and with *Six1/2>CD4::GFP* and *Six1/2>H2B::mCherry* reporter plasmids to visualize the OSP. Hatched larvae were fixed and imaged at 20 h postfertilization (hpf) and raised at 20°C. *Pcdh.e* CRISPR larvae were compared to negative control larvae electroporated instead with the standard *U6>Control* sgRNA plasmid that was designed to not target any *C. robusta* sequence and has been routinely used in the field (Stolfi et al., 2014).

CRISPR-mediated disruption of *Pcdh.e* appeared to result in loss of OSP compactness and separation from other surface ectoderm cells, compared to negative control larvae (Fig. 2A). OSP cells did not appear to form the typical rosette around a central canal connecting to the neural tube and endodermal lumen (Fig. 2B;

Fig. S2), as has been previously described (Veeman et al., 2010). We decided to quantify this loss of compactness by measuring the maximum length between anterior and posterior edges of the OSP, as visualized by *Six1/2>CD4::GFP* and *Six1/2>H2B::mCherry* reporter expression labeling cell membranes and nuclei, respectively. We measured a statistically significant expansion of the OSP surface area upon *Pcdh.e* CRISPR, and this effect was replicated in a duplicate experiment (Fig. 2C). These results suggested that *Pcdh.e* is necessary for maintenance of a cohesive, compact OSP territory in developing *Ciona* larvae. When we attempted to rear the larvae through metamorphosis and into the juvenile stage, nine of 17 *Pcdh.e* CRISPR juveniles had clearly deformed oral siphons, compared to none of ten negative *CRISPR* control juveniles (Fig. 2D). These deformities included larger-than-usual oral openings, disorganization of *Six1/2+* cells, and lack of clear separation between the oral siphon and anterior neural territory. Taken together, these data suggest that *Pcdh.e* expression in the OSP is crucial for its morphogenesis through embryogenesis and later post-metamorphic development into the oral siphon.

### Ectopic expression of Pcdh.e prevents proper OSP separation

Because we hypothesized that Pcdh.e is an adhesion molecule that maintains OSP integrity through homotypic cell-cell adhesion, we decided to overexpress Pcdh.e throughout the entire anterior neural plate using the *Foxc* promoter (*Foxc>Pcdh.e*) (Wagner and Levine, 2012). Foxc+ cells of the anterior neural plate will divide at the late gastrula stage to give rise to distinct OSP and papilla progenitors (Liu and Satou, 2019; Nicol and Meinertzhagen, 1988a,b; Wagner and Levine, 2012). Committed OSP and papilla progenitor cells are initially adjacent to one another at early tailbud stages; however, they slowly separate into two physically disjointed pools of Foxc+ cells separated by cells from other lineages not

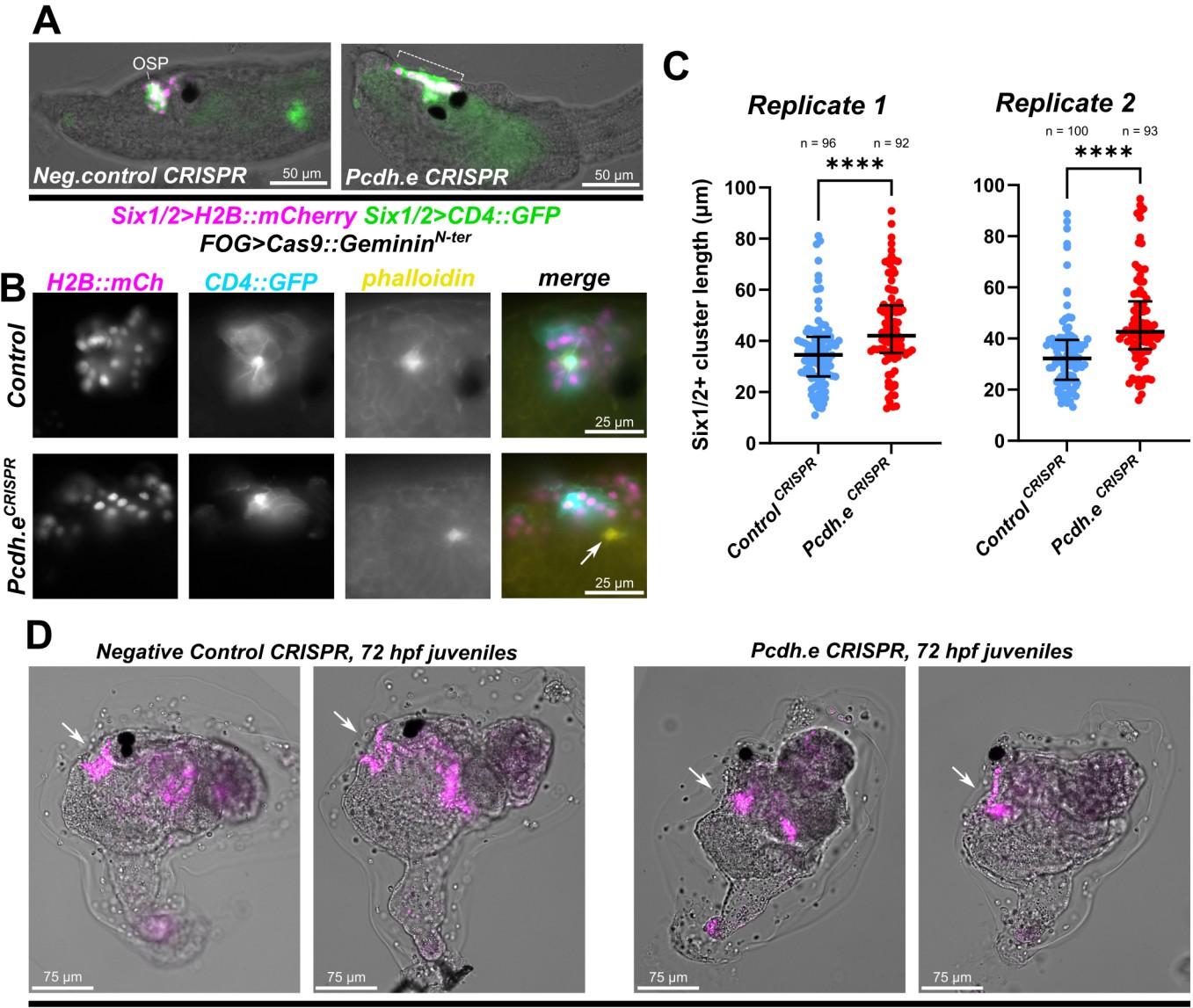

**Fig. 2. CRISPR-mediated mutagenesis of *Pcdh.e* disrupts OSP integrity and oral siphon development.** (A) Results of tissue-specific CRISPR/Cas9-mediated disruption of *Pcdh.e* in F0. Left: larva electroporated with *FOG>Cas9::Geminin^{N-ter}*, a negative control sgRNA, *Six1/2>H2B::mCherry*, and *Six1/2>CD4::GFP*. The *Six1/2* promoter drives OSP-specific expression of H2B::mCherry (magenta nuclei) and CD4::GFP (green cell membranes). The *FOG* promoter drives expression in animal pole lineages, including the lineage that gives rise to the OSP. Right: larva electroporated with the same components as in the left image, but with *Pcdh.e*-targeting sgRNAs (2.100 and 3.38) instead. The dashed bracket indicates aberrant OSP cell spreading anteriorly toward the papillae. Scale bars: 50 µm. (B) Higher-magnification images of representative OSPs in negative control (top) and *Pcdhe.e* CRISPR (bottom) larvae. OSP cell membranes were labeled with *Six1/2>CD4::GFP* (cyan in merged image), and nuclei were labeled with *Six1/2>H2B::mCherry* (magenta). Larvae counterstained with phalloidin-Alexa Fluor 647 (yellow). *Pcdh.e* CRISPR results in a more scattered, less compact OSP cell cluster. Notably, the cells do not form an opening connecting to the canal leading to the neural tube and endodermal lumen (white arrow). Negative control larva was electroporated with only the reporters, not Cas9 or sgRNA plasmids. Additional examples of larvae can be seen in Fig. S2. Scale bars: 25 µm. (C) Quantification of the anterior-posterior length of the Six1/2 reporter-expressing OSP cell territory in each imaged individual, compared between negative control and *Pcdh.e* CRISPR larvae. CRISPR was performed and analyzed in duplicate. ****$P<0.0001$ by a two-tailed Mann–Whitney *U* test. (D) Negative control (left) and *Pcdh.e* CRISPR (right) larvae were left to settle and undergo metamorphosis and imaged at 72 h postfertilization (hpf). Ten out of ten successfully metamorphosed negative control juveniles had no obvious oral siphon morphological defects, compared to nine out of 17 *Pcdh.e* CRISPR juveniles with clear defects. Oral siphon openings indicated by white arrows. Scale bars: 75 µm. See text for more details.

expressing Foxc (Veeman et al., 2010). As we expected, Pcdh.e overexpression by electroporation with a *Foxc>Pcdh.e* plasmid resulted in a single Foxc+ territory spread out over the entire anterior portion of the dorsal surface ectoderm (Fig. 3A,B). This suggested that the forced expression of Pcdh.e in normally Pcdh.e-negative papilla progenitors causes these cells to improperly associate

with (or prevents them from dissociating from) Pcdh.e+ OSP cells, resulting in the failure of the papilla and OSP to separate. In those larvae that successfully underwent metamorphosis and developed into juveniles, we found that 14 of 28 *Foxc>Pcdh.e*-electroporated animals also had clearly deformed oral siphons, with larger, less compact oral openings with *Foxc+* cells still abnormally spread out

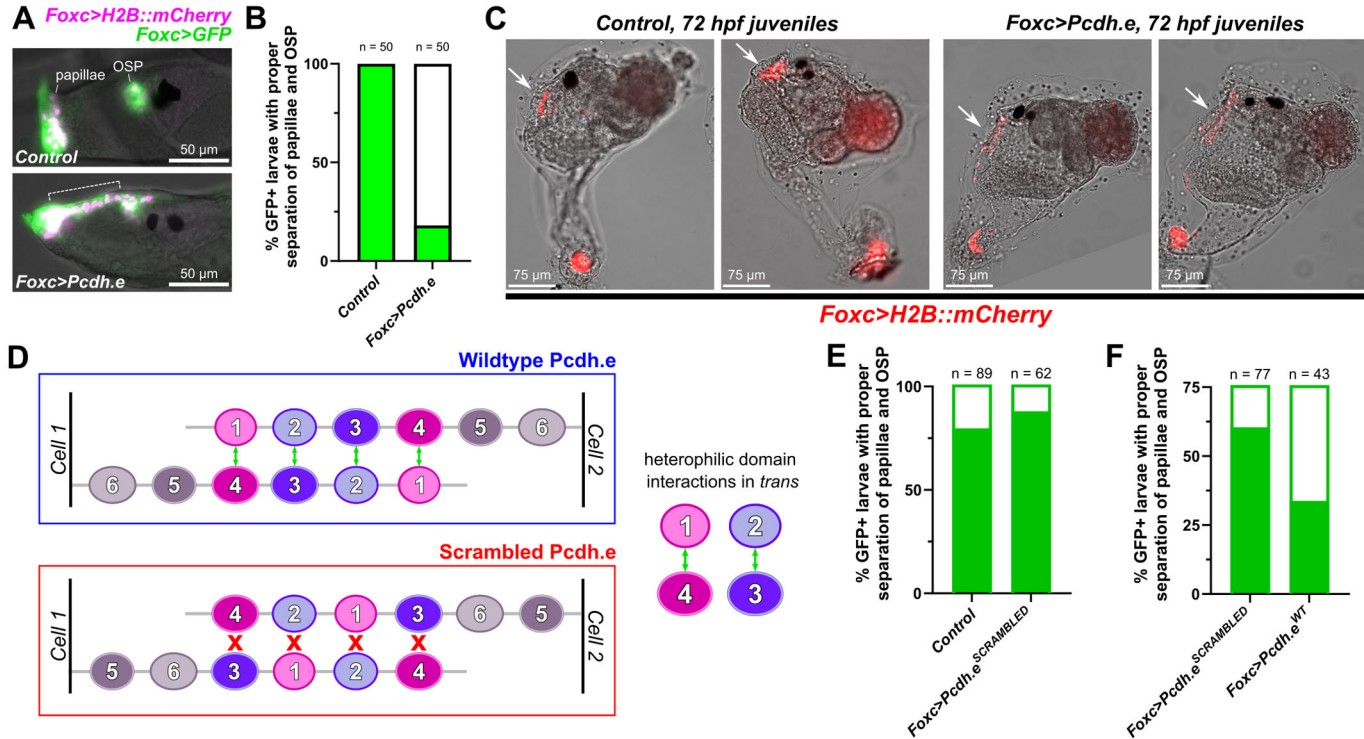

**Fig. 3. Precocious overexpression of Pcdh.e prevents the proper separation of the OSP from papilla progenitor cells.** (A) Top: wild-type/negative control larva electroporated only with *Foxc>H2B::mCherry* (magenta) and *Foxc>GFP* (green), revealing normal separation of the OSP territory and the papillae. Bottom: overexpression of Pcdh.e throughout the Foxc+ territory results in failure of OSP and papillae cells to properly segregate (dashed bracket). Scale bars: 50 µm. (B) Scoring the percentage of *Foxc>GFP*+ larvae with proper separation of OSP and papilla territories, showing a substantial decrease in the Pcdh.e overexpression condition. (C) Negative control (left) and *Pcdh.e* overexpression (*Foxc>Pcdh.e*; right) larvae underwent metamorphosis and were imaged at 72 hpf. Out of 45 control juveniles, 41 had normal-looking oral siphons, compared to only 14 of 28 in juveniles originally electroporated with *Foxc>Pcdh.e*. Oral siphon openings indicated by white arrows. Scale bars: 75 µm. See text for more details. (D) Top: diagram of the extracellular cadherin domains 1-6 of *Ciona robusta* wild-type Pcdh.e, with predicted heterophilic interactions *in trans* (green arrows) based on the determined structures of vertebrate protocadherins (explained in the diagram to the right). Bottom: diagram of the scrambled Pcdh.e molecule with the cadherin domains rearranged so as to disrupt antiparallel interactions *in trans*. Red Xs indicate predicted lack of heterophilic interaction between mismatched domains. Note that domains 5 and 6 are also swapped to help prevent possible interactions *in cis* between the scrambled Pcdh.e and endogenous, wild-type Pcdh.e. (E) Scoring the percentage of *Foxc>GFP*+ larvae with proper separation of OSP and papilla territories, assayed as in Fig. 2E,F, comparing the overexpression of the scrambled Pcdh.e (*Foxc>Pcdh.e^{SCRAMBLED}*) to a negative control in which the neutral reporter gene *lacZ* was overexpressed instead (control). No substantial difference in OSP-papilla separation was observed. (F) In contrast, a comparison between scrambled (*Foxc>Pcdh.e^{SCRAMBLED}*) and wild-type (*Foxc>Pcdh.e^{WT}*) Pcdh.e overexpression constructs revealed a substantial difference in perturbation of OSP-papilla separation, interpreted as being due to loss of heterophilic interactions *in trans* with the scrambled Pcdh.e. GFP in panel A tagged at N-terminus with Unc-76 tag (see Dataset 1).

between the oral siphon and the papilla-derived holdfast. This was in contrast to only four of 45 control animals showing any oral siphon defects (Fig. 3C). Our results suggest that Pcdh.e misexpression in surface ectoderm is sufficient to disrupt proper cell sorting and prevent normal separation between different placode-like structures of the *Ciona* larva, similarly to the *Pcdh.e* CRISPR knockout, with deleterious consequences for proper oral siphon formation in post-metamorphic juveniles.

### Pcdh.e function depends on canonical extracellular domain interactions *in trans*

Vertebrate protocadherins contain six extracellular calcium-binding cadherin domains (CA) that mediate cell-cell adhesion via domain-specific *trans* interactions in an antiparallel manner (Goodman et al., 2016; Rubinstein et al., 2015, 2017). According to this, we predicted that heterophilic interactions between CA1 and CA4 *in trans*, and between CA2 and CA3 (also *in trans*), are key to Pcdh.e function (Fig. 3D). We therefore created a 'scrambled' *Pcdh.e* cDNA, rearranging the CA domains to prevent such interactions *in trans*. Because domains CA5 and/or CA6 may mediate interactions *in cis* (Rubinstein et al., 2017), we

swapped these as well so as to lessen any potential dominant-negative effects.

When we overexpressed this scrambled Pcdh.e using the *Foxc* promoter (*Foxc>Pcdh.e^{SCRAMBLED}*), the papillae and OSP separated properly as in the negative control without any protocadherin overexpression (Fig. 3E). The overexpression of the scrambled Pcdh.e resulted in substantially fewer instances of fused papilla and OSP territories when directly compared to the wild-type Pcdh.e overexpression (Fig. 3F). This difference is not likely due to a difference in expression levels between the wild-type and the scrambled proteins, as both GFP-tagged variants were visibly expressed in larvae (Fig. S3A). In fact, Pcdh.eSCRAMBLED::GFP fluorescence was significantly higher than that of the wild-type Pcdh.e::GFP fusion, even when normalized to H2B::mCherry fluorescence (Fig. S3B). The reason for this is unclear, though the intracellular domain of vertebrate protocadherins is often cleaved and degraded, which may be regulated by extracellular interactions (Buchanan et al., 2010; Haas et al., 2005). Taken together, these data suggest that *Ciona* Pcdh.e functions like typical vertebrate protocadherins, which mediate cell-cell adhesion through their extracellular cadherin repeats.

## Six1/2 and Pitx transcription factors positively regulate *Pcdh.e* in the OSP

To further explore the regulatory pathways contributing to *Pcdh.e* expression in the OSP, we next investigated potential upstream transcription factors also important in vertebrate placode formation, Six1/2 and Pitx. Although their orthologs are conserved regulators of anterior placodes and placode-like structures in chordates (Schlosser et al., 2014), their roles in OSP formation in *Ciona* have never been investigated. *Six1/2* is expressed in the row of neural plate blastomeres that eventually give rise to the OSP (Imai et al., 2004; Mazet et al., 2005), while *Pitx* is expressed in the OSP at tailbud stages (Boorman and Shimeld, 2002; Christiaen et al., 2002).

To disrupt these genes, we designed and validated sgRNAs, targeting them for CRISPR knockout (Fig. S2B). We obtained one sgRNA targeting *Six1/2* at ~43% mutagenesis efficacy and two targeting *Pitx*, both measured to be at least 50% effective (Fig. S2B). To assay *Pcdh.e* expression in CRISPR larvae, we tested different fragments 5′ to the first exon (Fig. 4A). We found that a distal *cis*-regulatory sequence (*Pcdh.e −4608/−1447* relative to the start codon) was able to drive reporter gene expression in the OSP. A more proximal fragment (*Pcdh.e −1500/−1*) was not active in Foxc+ OSP cells but rather was expressed in adjacent cells just posterior to the OSP, likely the anterior apical trunk epidermal neurons (aATENs) based on their position (Fig. 4A) (Abitua et al., 2015). Animal pole-specific CRISPR-mediated disruption of *Six1/2* or *Pitx* resulted in modest and/or inconsistent loss of expression of the OSP-specific *Pcdh.e* reporter constructed using the −4608/−1447 fragment (*Pcdh.e[OSP]>GFP*) (Fig. 4B,C; Fig. S1). To better quantify this,

the mean fluorescence intensity of GFP expression was measured, normalized to mCherry co-expression, and compared between CRISPR and negative control larvae (Fig. 4D). This showed a significant reduction of *Pcdh.e[OSP]>GFP* expression in the OSP in either *Six1/2* or *Pitx* CRISPR larvae. These data show that, although neither Six1/2 nor Pitx might be absolutely required, they appear to contribute positively to *Pcdh.e* expression in the OSP.

## DISCUSSION

Here, using tissue-specific CRISPR/Cas9-mediated mutagenesis and gene overexpression, we present evidence that Pcdh.e is expressed in the OSP of the *Ciona* larva and is involved in its proper morphogenesis as a separate and cohesive group of surface ectoderm cells. CRISPR/Cas9-mediated mutagenesis of *Pcdh.e* in the ectoderm resulted in less cohesive OSP territory, as measured by its spreading over the surface of the larva. This is what one would expect if Pcdh.e is required for OSP cells to associate more closely with one another than to surrounding or related cells destined to give rise to separate structures like the papillae. In contrast, overexpression of Pcdh.e in the common progenitors of the OSP and papillae caused these two territories, which normally segregate from one another, to remain as a single cohesive unit. This abnormal fusion of OSP and papillae territories was not observed upon rearranging the cadherin repeats of Pcdh.e, suggesting that it functions through heterophilic interactions *in trans*, in a manner very similar to that of vertebrate protocadherins. We further link OSP-expressed transcription factors Six1/2 and Pitx to the transcriptional activity of a *Pcdh.e* reporter construct. Taken

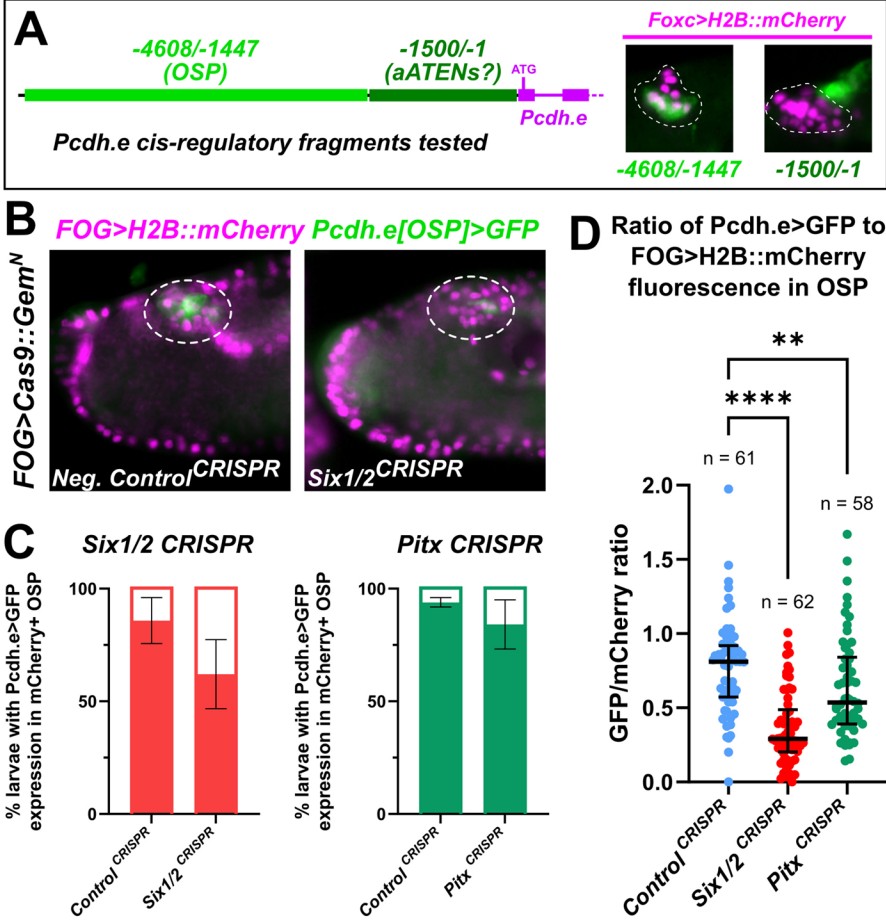

**Fig. 4. Regulation of *Pcdh.e* expression by OSP regulatory factors Six1/2 and Pitx.** (A) Left: diagram indicating distinct *cis*-regulatory sequences upstream of the *Pcdh.e* gene that drive reporter expression either in the OSP (−4608/−1447, henceforth referred to as *Pcdh.e[OSP]>GFP*) or in adjacent, more posterior cells that may correspond to aATENs (−1500/−1). Right: representative images of Unc-76::GFP reporter plasmids constructed using the indicated *cis*-regulatory elements, showing expression (green) in or near the OSP (dashed outline, labeled by Foxc>H2B::mCherry expression in magenta). See Dataset 1 for detailed sequence information. (B) Example of animal pole-lineage-specific CRISPR-mediated knockout of *Six1/2*, showing reduction of *Pcdh.e[OSP]>GFP* expression (green) in the OSP. Lineage marked by expression of *FOG>H2B::mCherry* (magenta). (C) Scoring of *Pcdh.e[OSP]>GFP* expression frequency in electroporated larvae, comparing *Six1/2* CRISPR and *Pitx* CRISPR conditions to negative controls. Sample size of at least 37 larvae (ranging up to 100) for each experiment, performed in duplicate. Error bars indicate the range of percentages. Individual duplicate graphs are shown in Fig. S4. One duplicate of the 'negative control' condition is shared between *Six1/2* and *Pitx* experiments, due to performing this set of electroporations in parallel. (C) Quantification of *Pcdh.e[OSP]>GFP/FOG>H2B::mCherry* fluorescence ratio in the OSPs of CRISPR and negative control larvae, showing significant reduction of *Pcdh.e* reporter expression upon disruption of *Six1/2* (****P<0.0001) and *Pitx* (**P=0.0072), by a two-tailed Mann–Whitney *U* test.

together, our results support a model in which differential expression of adhesion molecules in defined subsets of surface ectoderm cells allows for their segregation and self-organization into distinct embryonic territories, such as those that give rise to the mouth.

In vertebrates, classical cadherin adhesion molecules play crucial roles in the separation of placodes and the formation of cellular rosettes similar to that seen with the OSP of *Ciona* (Breau and Schneider-Maunoury, 2015). For instance, classical cadherins were found to be important for the separation and invagination of the lens placode in mice (Pontoriero et al., 2009). Furthermore, Cadherin2-rich apical junctions are important for rosette formation in zebrafish lateral-line primordia (Revenu et al., 2014). Our results suggest that regulation of cadherin superfamily-dependent cell-cell adhesion might be a conserved feature of placode development that predates the vertebrate-tunicate split.

In *Ciona*, there are two apparent 'classical' vertebrate-type cadherin genes (Sasakura et al., 2003), both of which are expressed in parts of the neural tube and surface ectoderm (Noda and Satoh, 2008). However, a higher diversity of expression patterns of the related protocadherins is seen during embryogenesis (Noda and Satoh, 2008). It was previously shown that delaminating tail neurons downregulate a distinct protocadherin gene (*Pcdh.c*) expressed in the surrounding surface ectoderm, and *Pcdh.c* mis-expression in these cells was sufficient to impair their delamination (Stolfi et al., 2015). This further suggests that multiple protocadherins can mediate crucial cellular adhesion and sorting functions in *Ciona*. We propose that the dynamic expression of protocadherin genes observed in the *Ciona* embryo might underlie many cell sorting and morphogenetic events, perhaps in concert with or complementing the function of classical cadherins.

While Pcdh.e might not be the only putative adhesion molecule to maintain the OSP as a distinct and cohesive rosette, our data suggest that it is a major factor in this morphogenetic process. In turn, the OSP gives rise to the future mouth (oral or incurrent siphon) of the post-metamorphic juvenile, later the adult. We show that perturbing the Pcdh.e function also affects the oral siphon of early juveniles. Additional structures and cell types arise later from this region, for instance, the velum and oral tentacles, which contain putative mechanosensory hair cell-like cells (Anselmi et al., 2024). Furthermore, other tissues are patterned around the mouth, such as the oral siphon muscles (Berrill, 1947; Chiba et al., 2004). Future studies will be required to see if OSP integrity, potentially mediated by Pcdh.e-dependent cell-cell adhesion, is required for the proper specification and morphogenesis of these post-metamorphic oral tissues and structures.

## MATERIALS AND METHODS
### *Ciona* handling, fixation, staining, and imaging
Adult *C. robusta* specimens were collected in California, around San Diego (M-REP) or Los Angeles/Orange County (Marinus Scientific). Dechorionated zygotes were generated and electroporated as described (Christiaen et al., 2009a,b). Embryos were raised in filtered and tris(hydroxymethyl)methylamino]propanesulfonic acid (TAPS)-buffered artificial seawater (FASW) at 20°C to desired Hotta stages (Hotta et al., 2020, 2007); then fixed and prepared onto microscope slides using a MEM-FA solution [3.7% formaldehyde, 0.1 M 3-(*N*-morpholino)propanesulfonic acid (MOPS) pH 7.4, 0.5 M NaCl, 1 mM ethylene glycol-bis(β-aminoethyl ether)-*N*,*N*,*N*′,*N*′-tetraacetic acid (EGTA), 2 mM $MgSO_4$, 0.1% Triton X-100]; rinsed in 1× PBS, 0.4% Triton X-100, 50 mM $NH_4Cl$ for autofluorescence quenching; and, finally, washed with 1× PBS, 0.1% Triton X-100. Phalloidin-Alexa Fluor 546 or 647 (Thermo Fisher Scientific) staining was carried out using a 1:50 dilution, as previously described

(Lowe et al., 2021). Imaging was accomplished using Leica DMi8 or DM IL LED inverted epifluorescence and scanning point confocal microscopes.

### Fluorescence intensity quantification
For fluorescence quantification, larvae were imaged on a Leica DMi8 using constant exposure for GFP and mCherry channel images, and mean fluorescence intensity in regions of interest encompassing mCherry+/GFP+ OSP and/or papilla territories (the latter included in the quantification of Pcdh.e::GFP fusions) was recorded. GFP-to-mCherry mean fluorescence intensity ratios were calculated for each individual region of interest in order to normalize potential discrepancies in electroporation efficiency (Shimai and Veeman, 2021). One outlier (ratio>2.0) in the control and one in the *Pitx* CRISPR condition were removed. Statistical data were plotted and analyzed in GraphPad Prism. Previously published and new plasmid sequences can be found in Dataset 1. All plasmids are available upon request.

### CRISPR/Cas9 sgRNA design and validation
sgRNAs were designed using CRISPOR (http://crispor.tefor.net/), and expression vectors were constructed as previously described (Gandhi et al., 2017, 2018) or custom synthesized and cloned (Twist Bioscience). 75 µg of each individual sgRNA vector was validated *in vivo* as previously described (Popsuj et al., 2024) by co-electroporation with 25 µg of a ubiquitous (Sasakura et al., 2010) *Eef1a* −1955/−1>*Cas9* or *Eef1a* −1955/−1>*Cas9::Geminin* plasmid (per 700 µl of total electroporation volume). Genomic DNA was isolated using a QIAamp Extraction Micro Kit (QIAGEN), targeted regions were amplified by PCR using AccuPrime Pfx (Thermo Fisher Scientific), and PCR products were purified using a QIAquick PCR Purification Kit (QIAGEN) following the published protocol (Johnson et al., 2023). Amplicons were sequenced using Illumina-based amplicon sequencing (Amplicon-EZ; Azenta), and efficiency was determined by indel plotting as automatically generated by Azenta. In total, four sgRNAs targeting *Pcdh.e* were created and validated, and two were selected for further use: sgRNAs 2.100 and 3.38, targeting exons 2 and 3, respectively. For *Six1/2* CRISPR, three sgRNAs targeting the first exon were created and validated, and one was selected for further use: sgRNA 1.358. For *Pitx* CRISPR, two targeting the second constitutive exon were created and validated, and both were used: sgRNAs 2.126 and 1.186. The predicted protein domain organization of Pcdh.e was analyzed by SMART (Letunic et al., 2021).

### Tissue-specific CRISPR/Cas9 and overexpression of *Pcdh.e*
*Pcdh.e* was specifically disrupted by CRISPR/Cas9 in the animal pole lineages using 25 µg of *FOG>Cas9::Geminin-Nter* (Pennati et al., 2024; Song et al., 2022) co-electroporated with 45 µg of *U6>Pcdh.e.2.100* sgRNA plasmid, 45 µg of *U6>Pcdh.e.3.38* sgRNA plasmid, 50 µg of *Six1/2>CD4::GFP*, and 25 µg of *Six1/2>H2B::mCherry*. Results were compared to co-electroporation with 90 µg of a standard 'negative control' sgRNA (not targeting any known *C. robusta* genomic sequence) vector instead, as previously published (Stolfi et al., 2014). The *Six1/2* promoter used was based on previously cloned and published promoters (Abitua et al., 2015). The *FOG* (*Friend of GATA*, e.g. *Zfpm*, KyotoHoya ID KH.C10.536) promoter has been published and is extensively used to drive Cas9 in animal pole lineages (Pennati et al., 2024; Rothbächer et al., 2007). Distances and lengths were measured in Leica LAS X software and statistically analyzed and plotted in GraphPad Prism. We manually excluded outliers >100 µm (fewer than ten individuals in any one condition) as these could not be confidently distinguished from cases of leaky reporter expression in more anterior cells or other nonspecific developmental defects. To overexpress *Pcdh.e* in the anterior neural plate (common progenitors of the OSP+ papilla territory), we used the published *Foxc* promoter (Wagner and Levine, 2012), co-electroporating 90 µg of *Foxc>Pcdh.e* (wild type or 'scrambled', fused or not to GFP), 25 µg of *Foxc>H2B::mCherry*, and 50 µg of *Foxc>Unc-76::GFP*. This was compared to a negative control without *Foxc>Pcdh.e* or with the neutral *Foxc>lacZ* nonfluorescent reporter plasmid in place of the overexpression plasmid. New and previously published plasmid sequences can be found in Dataset 1.

### Tissue-specific CRISPR/Cas9 of *Six1/2* and *Pitx*
*Six1/2* and *Pitx* genes were specifically disrupted by CRISPR/Cas9 in the animal pole lineages using 30 µg of *FOG>Cas9::Geminin-Nter*

co-electroporated with 90 µg of *U6>Six1/2.1.358*, 45 µg each of *U6>Pitx.2.126* and *U6>Pitx.2.186*, or 90 µg of 'negative control' sgRNA plasmids. All were co-electroporated with 80 µg of *Pcdh.e −4608/−1447+ bpFOG>Unc-76::GFP* and 25 µg of *FOG>H2B::mCherry* reporter plasmids. New and previously published plasmid sequences can be found in Dataset 1.

## Raising juveniles

To raise animals to the desired juvenile stage, larvae were initially raised until 19-22 hpf on 60×15 mm Petri dishes coated with 1% agarose in FASW. Larvae were then transferred into new 100×15 mm uncoated Petri dishes filled with ∼30 ml of FASW and 100 µl of 100× penicillin-streptomycin (Gibco). Twice daily, the FASW was exchanged two times in quick succession: FASW was completely poured off before being quickly but carefully replaced with ∼30 ml of new FASW. After 15 s, that FASW was also poured off and replaced with another ∼30 ml of new FASW, with penicillin-streptomycin added as described before. Juveniles were raised at 20°C in a controlled-temperature incubator to their desired stage (72 hpf). For collection, juveniles were first paralyzed in FASW containing 100 µg/ml of tricaine mesylate (Sigma-Aldrich) for 15 min before being carefully scraped off with thin forceps, followed by fixation as described above for larvae.

## Acknowledgements

We thank Susanne Gibboney, Lindsey Cohen, and Wesley Bartlett for technical assistance, as well as other members of the lab for their advice. We thank Gwynna Fuller, Ankita Thawani, Andy Groves, and Christina Cota for their helpful feedback.

## Competing interests

The authors declare no competing or financial interests.

## Author contributions

Conceptualization: S.V., B.J., A.S., S.P.; Data curation: A.S.; Formal analysis: S.V., B.J., A.S., S.P.; Funding acquisition: S.V., A.S., S.P.; Investigation: S.V., B.J., C.J.J.; Project administration: A.S., S.P.; Resources: B.J.; Supervision: C.J.J., A.S., S.P.; Writing – original draft: S.V., A.S.; Writing – review & editing: B.J., S.P.

## Funding

S.V. was funded by a PURA scholarship award from Georgia Institute of Technology. S.P. was funded by Achievement Rewards for College Scientists Foundation and Mortar Board fellowships. This work was funded by grant R01HD104825 from the Eunice Kennedy Shriver National Institute of Child Health and Human Development, grant R35GM158421 from the National Institute of General Medical Sciences, and grant 1940743 from the Division of Integrative Organismal Systems (National Science Foundation). Open Access funding provided by Georgia Institute of Technology. Deposited in PMC for immediate release.

## Data and resource availability

All relevant data and details of resources can be found within the article and its supplementary information.

## First Person

This article has an associated First Person interview with the joint first authors of the paper.

## Peer review history

The peer review history is available online at https://journals.biologists.com/bio/lookup/doi/10.1242/bio.062169.reviewer-comments.pdf

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
