## [Peer Review File · Biology Open]

A protocadherin mediates oral placode morphogenesis in the tunicate *Ciona*

Sriikhar Vedurupaka, Bitu Jadali, Christopher J. Johnson, Alberto Stolfi and Sydney Popsuj
DOI: 10.1242/bio.062169

Editor: Tristan Rodríguez

Review timeline

Original submission:	18 July 2025
Editorial decision:	25 July 2025
First revision received:	5 January 2026
Accepted:	12 January 2026

Original submission

First decision letter

MS ID#: bio.062169

MS Title: A protocadherin mediates oral placode morphogenesis in the tunicate *Ciona*

Authors: Sriikhar Vedurupaka, Bitu Jadali, Christopher J. Johnson, Alberto Stolfi and Sydney Popsuj

I am writing to let you know that I have now reached a decision on the above manuscript. I am afraid that, after careful consideration, I feel that it cannot currently be accepted for publication in Biology Open.

The reviewer reports are shown at the bottom of this email or can be accessed, together with a copy of this decision letter, by going to:

As you will see, the reviewers raise a number of substantial criticisms that prevent me from accepting your paper for publication.

I realise that this is disappointing news, and we understand the frustration that you must feel. However, I am sure that you appreciate that the conclusions of your research must be seen by the wider community to be fully supported by the data. On this occasion, I have decided that this is not the case.

Reviewer 1

Comments for the author

In this manuscript, Stolfi, Popsuj and colleagues use *Ciona robusta*, a model system to address chordate evolution and the origin of vertebrates, to characterize the role of differential cell adhesion in the development of the mouth primordium, a cranial like placode termed oral siphon placode (OSP). The paper is subdivided into three different chapters or figures. In the first figure, authors identify Protocadherin-e (among the six existing protocadherins present in *Ciona*) as the one whose expression is restricted to the OSP. Unfortunately, the expression of the remaining protocadherins is not mentioned as to whether they are also implicated in OSP development. In the second figures, authors use CRISPR/Cas9 technology and overexpression experiments to functionally propose a role of Protocadherin-e in driving the cohesion of those cells that constitute the OSP,

monitored by the length of the structure. Whether other more classical ways of measuring cohesion based on differential cell adhesion with surrounding cells (eg. circularity) can be used to more efficiently demonstrate the implication of this protein in cohesion is suggested by this reviewer. Moreover, a more detailed characterization of the phenotype by state-of-the-art microscopy techniques (and not just GFP epifluorescence) is required to consolidate their conclusions. Whether the experimental conditions have an effect on overall tissue morphogenesis and not just OSP cohesion is an open possibility that cannot be discarded by their experimental design and results. Thus, their results are suggestive but not conclusive. In the third figure, authors identify a minimal cis-regulatory sequence that drives expression of Protocadherin-e to the OSP and conclude that the transcription factors Six1/2 and Pitx contribute to the restricted expression of Protocadherin-e to the OSP. It is unclear how and why they normalized the expression of the GFP reporter to the expression of FOG-mCherry in these experiments. In the case they do not do it, no significant differences between control and experimental conditions are observed. Overall, and based on the comments listed above, conclusions are not convincing, experimental data require better quantification and deeper analysis, and the paper sounds highly preliminary in its present form.

Reviewer 2

Comments for the author

The authors have done amazing work on investigating the molecular basis for the formation and structural cohesion of the oral siphon placode (OSP) in the tunicate *Ciona robusta*, a basal chordate model. The study provides molecular evidence for the conservation of placodal mechanisms outside vertebrates. They also propose that differential cell adhesion, mediated by protocadherins, is a primitive and conserved feature of sensory placode development and reinforces the utility of *Ciona* as a model for studying chordate evolution and neurodevelopmental patterning.

I have few questions:

- 1) Functional redundancy of protocadherins is not fully explored. Could additional genes compensate for loss of Pcdh.e.
- 2) Why long-term phenotypic consequences of Pcdh.e disruption (e.g., in the adult oral siphon) were not assessed.
- 3) Figures showing expression in live or fixed embryos (e.g., fluorescence) should be supplemented with higher-resolution insets, especially to highlight cell rosette disruption (Example Fig 2 and 3).
- 4) The cellular mechanism by which Pcdh.e mediates adhesion is not explored.

This is a strong, well-executed study that significantly enhances our understanding of developmental mechanisms in basal chordates and their evolutionary relevance. I recommend publication pending minor revisions focused on enhancing mechanistic clarity and providing additional visualization of results.

Reviewer's Responses to Questions

Experimental quality

Does each figure have the proper controls?

If 'No', please indicate reasons in Comments for Author box below.

Reviewer #1:

- Yes

Reviewer #2:

- Yes

Were the data analyzed using appropriate statistical tests?

If 'No', please indicate reasons in Comments for Author box below.

Reviewer #1:

- Yes

Reviewer #2:

- Yes

Reproducibility

Were experiments performed using adequate number of biological replicates?

If 'No', please indicate reasons in Comments for Author box below.

Reviewer #1:

- Yes

Reviewer #2:

- Yes

Does the methods section provide sufficient detail to permit reproducibility?

If 'No', please indicate reasons in Comments for Author box below.

Reviewer #1:

- No

Reviewer #2:

- Yes

Completeness

Are the manuscript's conclusions supported by the data?

If 'No', please indicate reasons in Comments for Author box below.

Reviewer #1:

- No

Reviewer #2:

- Yes

Scholarship

Do the authors cite and discuss the merits of data that would argue for and against their conclusion?

If 'No', please indicate reasons in Comments for Author box below.

Reviewer #1:

- Yes

Reviewer #2:

- Yes

Does the manuscript title & abstract accurately reflect the contents of the manuscript, without hyperbole?

If 'No', please indicate reasons in Comments for Author box below.

Reviewer #1:

- No

Reviewer #2:

- Yes

First revision

Author response to reviewers' comments

We thank the reviewers for their critical feedback on our original submission. See below our © 2026. Published by The Company of Biologists under the terms of the Creative Commons Attribution License (<https://creativecommons.org/licenses/by/4.0/>).

brief point-by-point replies to the reviews:

Reviewer 1:

In the first figure, authors identify Prododcadherin-e (among the six existing protocadherins present in *Ciona*) as the one whose expression is restricted to the OSP. Unfortunately, the expression of the remaining protocadherins is not mentioned as to whether they are also implicated in OSP development

We originally cited the paper that comprehensively documented the expression of all protocadherins during *Ciona* development. As we now clarify in the revised manuscript, *Pcdh.e* is the only one that is expressed exclusively in the oral siphon placode cells within the surface ectoderm (though it is also expressed in a motor ganglion neuron subtype further posterior and internally in the neural tube).

In the second figures, authors use CRISPR/Cas9 technology and overexpression experiments to functionally proposes a role of Prododcadherin-e in driving the cohesion of those cells that constitute the OSP, monitored by the length of the structure. Whether other more classical ways of measuring cohesion based on differential cell adhesion with surrounding cells (eg. circularity) can be used to more efficiently demonstrate the implication of this protein in cohesion is suggested by this reviewer. Moreover, a more detailed characterization of the phenotype by state-of-the-art microscopy techniques (and not just GFP epifluorescence) is required to consolidate their conclusions. Whether the experimental conditions have an effect on overall tissue morphogenesis and not just OSP cohesion is an open possibility that cannot be discarded by their experimental design and results. Thus, their results are suggestive but not conclusive.

Our work is intended to be firmly developmental in scope, and we have therefore adjusted our claims and even our title (now titled “*A protocadherin mediates oral placode morphogenesis in the tunicate Ciona*”) to reflect our focus on OSP morphogenesis and oral siphon development. We have nonetheless added higher-resolution images of the phenotypes, using membrane-bound GFP and phalloidin staining to capture the morphological defects we see (Figures 2B and S2). We agree that state-of-the-art microscopy will be needed to characterize the phenotype at a subcellular level, though we believe this is beyond the scope of our intended focus.

On the question of *Pcdh.e* function, we have now added data showing that a domain-scrambled *Pcdh.e* does not induce OSP and papilla cohesion the same way that wild-type *Pcdh.e* does (Figure 3D-F), which is consistent with this protein’s proposed function in cell-cell adhesion through heterophilic binding between the different extracellular domains (arranged in an antiparallel manner *in trans* between two opposing cells). Using GFP-tagged versions of the wild-type and domain-scrambled *Pcdh.e* (Figure S3), we also show that the failure of the latter to induce improper cohesion (leading to a defect in the separation of OSP and papilla territories) is not due to reduced expression, as quantitative measurements showed that, if anything, the scrambled protein is expressed at higher levels than the wild type protein. Therefore, we do believe there is a connection between OSP morphogenesis and the known molecular mechanisms of protocadherin function.

In the third figure, authors identify a minimal cis-regulatory sequence that drives expression of Prododcadherin-e to the OSP and conclude that the transcription factors Six1/2 and Pitx contribute to the restricted expression of Prododcadherin-e to the OSP. It is unclear how and why they normalized the expression of the GFP reporter to the expression of FOG-mCherry in these experiments. In the case they do not do it, no significant differences between control and experimental conditions are observed.

The ratios between GFP and mCherry signals (dot plots, now in Figure 4D of the revised paper) represent a normalization or electroporation efficacy. We make this clearer now, in the methods and in the results, citing also Shimai and Veeman who formally described this normalization method. Based on this, we believe there are significant differences

Reviewer 2:

1) Functional redundancy of protocadherins is not fully explored. Could additional genes

compensate for loss of Pcdh.e.

While redundancy is a possibility, it needs not be invoked as we see a major defect upon knocking out *Pcdh.e* alone. Combined with the fact that no other protocadherin gene is enriched in the OSP relative to other nearby surface ectoderm cells (a point now made more clearly in our revised manuscript), we believe this exploration is not an immediate priority.

2) Why long-term phenotypic consequences of *Pcdh.e* disruption (e.g., in the adult oral siphon) were not assessed.

In our revised manuscript we have now followed some *Pcdh.e* CRISPR and overexpression animals through metamorphosis and show clear defects in oral siphon morphology that is consistent with the effects we report in the corresponding larvae (Figures 2D and 3C). The sample sizes are relatively small (reported in the respective figure legends), due to the fact that raising juveniles, especially those that have been genetically manipulated, can be difficult and prone to specimen quality issues beyond our control.

3) Figures showing expression in live or fixed embryos (e.g., fluorescence) should be supplemented with higher-resolution insets, especially to highlight cell rosette disruption (Example Fig 2 and 3).

We now have included higher-resolution images for the CRISPR phenotype (Figures 2B and S2). However, as we mention in our reply to reviewer 1, we have stepped back from making claims at the sub-cellular level and re-focus our paper's scope on the developmental outcomes, at the tissue/territory level.

4) The cellular mechanism by which *Pcdh.e* mediates adhesion is not explored.

We have addressed this now with the overexpression of a domain-scrambled *Pcdh.e* variant. Based on structural analysis of protocadherin binding in vertebrates, they do so *in trans* through an anti-parallel configuration, relying on heterophilic binding between different extracellular domains. By scrambling the order of these domains in *Pcdh.e*, we predicted that its ability to mediate cell-cell adhesion would be lost. Indeed, overexpression of a scrambled *Pcdh.e* did not cause significantly aberrant OSP/papilla morphogenesis, the way that wildtype *Pcdh.e* clearly does (Figure 3D-F). Using GFP-tagged versions, we show that this difference is likely not due to lower expression/folding (Figure S3). We conclude that *Pcdh.e* is likely to function similarly to vertebrate protocadherins, which have been extensively studied.

This is a strong, well-executed study that significantly enhances our understanding of developmental mechanisms in basal chordates and their evolutionary relevance. I recommend publication pending minor revisions focused on enhancing mechanistic clarity and providing additional visualization of results.

We thank the reviewer for their helpful feedback and support.

Second decision letter

MS ID#: bio.062169R1

MS Title: A protocadherin mediates oral placode morphogenesis in the tunicate *Ciona*

Authors: Sriikhar Vedurupaka, Bitu Jadali, Christopher J. Johnson, Alberto Stolfi and Sydney Popsuj

I am happy to tell you that your manuscript has been accepted for publication in Biology Open, pending our standard publication integrity checks. It was accepted on 12th January 2026.